# Insulin-like Growth Factor 1 Promotes Cell Proliferation by Downregulation of G-Protein-Coupled Receptor 17 Expression via PI3K/Akt/FoxO1 Signaling in SK-N-SH Cells

**DOI:** 10.3390/ijms23031513

**Published:** 2022-01-28

**Authors:** Ka-Na Lin, Kan Zhang, Wei Zhao, Shi-Ying Huang, Hao Li

**Affiliations:** 1Center for Brain Science & Clinical Research Center, Shanghai Children’s Medical Center, School of Medicine, Shanghai Jiao Tong University, Shanghai 200127, China; linkana@scmc.com.cn; 2Department of Anesthesiology, Shanghai Children’s Medical Center, School of Medicine, Shanghai Jiao Tong University, Shanghai 200127, China; zhangkan@scmc.com.cn; 3Department of Pharmacy, Shanghai Children’s Medical Center, School of Medicine, Shanghai Jiao Tong University, Shanghai 200127, China; zhaowei@scmc.com.cn (W.Z.); huangshiying@scmc.com.cn (S.-Y.H.)

**Keywords:** insulin-like growth factor 1, G-protein-coupled receptor 17, FoxO1, PI3K, Akt, cell damage, ischemic stroke

## Abstract

Insulin-like growth factor 1 (IGF-1) not only regulates neuronal function and development but also is neuroprotective in the setting of acute ischemic stroke. G-protein-coupled receptor 17 (GPR17) expression in brain tissue serves as an indicator of brain damage. As whether IGF-1 regulates GPR17 expression remains unknown, the aim of this study is to investigate how IGF-1 regulates GPR17 expression in vitro. Human neuroblastoma SK-N-SH cells were used. Lentivirus-mediated short hairpin RNA (shRNA) was constructed to mediate the silencing of FoxO1, while adenoviral vectors were used for its overexpression. Verification of the relevant signaling cascade was performed using a FoxO1 inhibitor (AS1842856), a phosphatidylinositol 3-kinase (PI3K) inhibitor (LY294002), and a GPR17 antagonist (cangrelor). Cell proliferation was analyzed using EdU staining; immunofluorescence staining was used to detect the expression and subcellular localization of FoxO1. Chromatin immunoprecipitation was used to analyze the binding of FoxO1 to the GPR17 promoter in SK-N-SH cells. The expression of FoxO1, GPR17, and protein kinase B (also known as Akt) mRNA and protein as well as the levels of FoxO1 and Akt phosphorylation were investigated in this study. IGF-1 was found to downregulate FoxO1 and GPR17 expression in SK-N-SH cells while promoting cell viability and proliferation. Inhibition of FoxO1 and antagonism of GPR17 were found to play a role similar to that of IGF-1. Silencing of FoxO1 by lentivirus-mediated shRNA resulted in the downregulation of FoxO1 and GPR17 expression. The overexpression of FoxO1 via adenoviral vectors resulted in the upregulation of FoxO1 and GPR17 expression. Blocking of PI3K signaling by LY294002 inhibited the effect of IGF-1 on GPR17 suppression. Results from chromatin immunoprecipitation revealed that IGF-1 promotes FoxO1 nuclear export and reduces FoxO1 binding to the GPR17 promoter in SK-N-SH cells. Here, we conclude that IGF-1 enhances cell viability and proliferation in SK-N-SH cells via the promotion of FoxO1 nuclear export and reduction of FoxO1 binding to the GPR17 promoter via PI3K/Akt signaling. Our findings suggest that the enhancement of IGF-1 signaling to antagonize GPR17 serves as a potential therapeutic strategy in the management of acute ischemic stroke.

## 1. Introduction

Acute ischemic stroke (AIS) is defined as sudden focal ischemia of brain tissue lasting for over 24 h and resulting in neurologic dysfunction [1]. Numerous conditions, such as embolism, atherosclerotic disease, vessel dissection, and vasculitis, lead to focal cerebral hypoperfusion, thus causing AIS [1]. Thrombolytic and antithrombotic agents are typically used in the management of AIS [2]. Neuroprotection and stem cell therapy, however, are emerging strategies relevant to AIS management that aim for early recovery of neurologic function [3]. As such, strategies to promote neuronal regeneration warrant investigation.

IGF-1 is a pleiotropic peptide mainly secreted from the liver that regulates both cellular function and development [4]. IGF-1 can be produced by all neuronal cell types and functions in both autocrine and paracrine manners. Furthermore, it possesses neuroprotective, neuroregenerative, and vasculoprotective properties following brain injury [5,6,7]. Two major signaling pathways, the PI3K/Akt and Ras/ERK cascades, are activated by IGF-1 [4]. Importantly, Akt acts as a critical mediator of growth-factor-induced neuronal survival [8,9]. PI3K/Akt-cascade-induced phosphorylation of forkhead box protein O1 (FoxO1) results in its nuclear importation and promotes neuronal growth and survival [10,11,12].

GPR17 is primarily expressed in the ischemic brain, heart, and kidneys and has been described as an orphan receptor phylogenetically located at an intermediate position between the P2Y and CysLT receptors [13]. Numerous studies have reported GPR17 to be a sensor of brain damage, playing a crucial role in lesion remodeling and repair [14,15,16]. This receptor is also known to be a key regulator in the maturation of oligodendrocyte precursors [17,18]. Antagonism of GPR17 was reported to attenuate brain tissue damage, suggesting that GPR17 is a potential pharmacological target for the treatment of nervous system diseases, such as ischemia [16,19]. Interestingly, GPR17 was identified as an effector of FoxO1 orexigenic signals in agouti-related peptide neurons [20,21,22]. However, the relationship between FoxO1 and GPR17 and the effect of FoxO1 on neuronal survival remain unclear. To mimic the nerve injury under AIS, SK-N-SH cells were starved for 24 h. Here, we investigate the influence of IGF-1 on neuronal GPR17 expression as well as the underlying molecular mechanisms of this interaction.

## 2. Results

### 2.1. IGF-1 Downregulates FoxO1 and GPR17 Expression in SK-N-SH Cells

To explore the influence of IGF-1 on FoxO1 and GPR17 expression, SK-N-SH cells were evaluated after exposure to IGF-1 at different concentrations within 24 h; 100 ng/mL of IGF-1 was found to significantly (*p* < 0.01) downregulate the mRNA expression of both *FOXO1* and *GPR17* after 1 h, and such downregulation lasted for at least 24 h (Figure 1a), suggesting a negative regulatory effect of IGF-1 on FoxO1 and GPR17 transcription. Results from Western blot (WB) analysis revealed that while IGF-1 downregulated FoxO1 expression, GPR17 expression was upregulated in the first 3 h and only subsequently downregulated (Figure 1b,c and Appendix A). Regardless of the IGF-1 concentration (0, 25, 50, 100, and 200 ng/mL), IGF-1 was found to significantly (*p* < 0.01) reduce *FOXO1* and *GPR17* mRNA expression in SK-N-SH cells after 24 h of exposure (Figure 1d). Cellular protein levels of both FoxO1 and GPR17 were found to be downregulated after exposure to IGF-1 for 24 h (Figure 1e,f and Appendix A).

### 2.2. Blocking FOXO1 and GPR17 Promotes SK-N-SH Cell Proliferation

To further investigate the interrelationship of IGF-1, FoxO1, and GPR17, cell viability and proliferation were evaluated after SK-N-SH cells were treated with IGF-1, a FoxO1 inhibitor (AS1842856) and a GPR17 antagonist (cangrelor). IGF-1 (25, 50, 100, and 200 ng/mL), AS1842856 (50 and 100 nM), and cangrelor (5 and 10 μM) were found to significantly increase cellular viability (*p* < 0.01; Figure 2a). IGF-1 (100 ng/mL) was found to significantly increase EdU-positive staining (EdU+) of cells, indicating that IGF-1 promotes SK-N-SH cell proliferation (Figure 2b). With the blockade of FoxO1 by AS1842856 and GPR17 by cangrelor, the number of EdU+ SK-N-SH cells also increased (Figure 2b). Representative images of EdU+ SK-N-SH cells treated with or without IGF-1, AS1842856, and cangrelor are shown in Figure 2c. We conclude that IGF-1 promotes SK-N-SH cell proliferation by inhibiting *FOXO1* and *GPR17* expression.

### 2.3. FoxO1 Regulates GPR17 Expression

To further confirm the influence of FoxO1 on GPR17, we used packaged lentivirus-mediated shRNA to reduce *FOXO1* expression and packaged adenovirus to promote *FOXO1* expression. Lentivirus-mediated shRNA was found to significantly downregulate the mRNA expression of *FOXO1* and *GPR17* (*p* < 0.01; Figure 3a), indicating that FoxO1 regulates the basal expression of GPR17. WB analysis revealed that lentivirus-mediated shRNA significantly downregulates the protein expression of FoxO1 (*p* < 0.01) but not of GPR17 (Figure 3b,c and Appendix A). The use of adenoviral packaging significantly increased the mRNA expression of *FOXO1* (*p* < 0.01; Figure 3d). The mRNA expression of *GPR17* was found to be upregulated with the increase in Ad-Foxo1-AAA; levels were noted to be significantly elevated when the titer of Ad-FoxO1-AAA surpassed 100 MOI (Figure 3d). WB analysis revealed only FoxO1 expression to have been significantly increased (Figure 3e,f and Appendix A). These results indicated that FoxO1 regulates the basal expression of GPR17 and that the knockdown of FoxO1 results in the downregulation of GRP17.

### 2.4. IGF-1 Influences the PI3K/Akt/FoxO1 Signaling Cascade and Thus GPR17 Expression in SK-N-SH Cells

To further explore how IGF-1 downregulates GPR17, we studied the PI3K/Akt/FoxO1 signaling cascade in SK-N-SH cells. WB analysis revealed pAkt and pFoxo1 to be significantly upregulated after treatment with 100 ng/mL of IGF-1 for 5 min (Figure 4a–c). However, Akt and FoxO1 expression was not affected by IGF-1 within a period of 60 min of treatment. In addition, LY294002 was used to block this signaling pathway. Interestingly, although LY294002 was noted to reverse the downregulatory effect of IGF-1 on *FOXO1* mRNA expression, only a slight reversal of *GPR17* mRNA expression was found (Figure 4d). Results from WB analysis indicated that blocking the PI3K/Akt/FoxO1 signaling pathway results in the effects of IGF-1 on PI3K/Akt/FoxO1 signaling to be muted (Figure 4e,f,c). However, WB findings revealed that GPR17 protein expression is less affected by IGF-1 and LY294002, and no significant difference was found (Appendix A). These results indicate that IGF-1 governs GPR17 expression via PI3K/Akt/FoxO1 signaling in SK-N-SH cells.

### 2.5. IGF-1 Promotes FoxO1 Nuclear Export and Reduces the Binding of FoxO1 to the GPR17 Promoter in SK-N-SH Cells

To investigate the role of FoxO1 in IGF-1-induced GPR17 gene transcription, cell viability and proliferation were analyzed after SK-N-SH cell treatment with IGF-1 and with or without LY294002. Importantly, LY294002 was found to inhibit the cell viability and reverse the effects of IGF-1 (Figure 5a). LY294002-mediated blocking of the PI3K-AKT-FoxO1 signaling cascade significantly reduced EdU+ SK-N-SH cells (*p* < 0.01) and reversed the effects of IGF-1 on cellular proliferation (Figure 5b). Representative images of EdU+ SK-N-SH cells with or without IGF-1 and LY294002 are shown in Figure 5c.

Moreover, FoxO1 staining with or without IGF-1 and LY294002 was analyzed. As shown in Figure 5d, IGF-1 promoted FoxO1 nuclear export and reduced the binding of FoxO1 to the GPR17 promoter. Results from ChIP assays revealed that 100 ng/mL of IGF-1 significantly reduced the binding of FoxO1 to the GPR17 promoter regions in SK-N-SH cells (Figure 5e). However, we did not observe an enriched binding of FoxO1 to GPR17 promoter when using the adenoviral FoxO1-AAA overexpression vectors (Figure 5e,f). This is likely because the time point we collected the cells for ChIP assay is too early and the overexpression of FoxO1 is not fully achieved (Figure 5f).

These results indicate that IGF-1 promotes FoxO1 nuclear export and reduces its binding to the GPR17 promoter regions in SK-N-SH cells, thus resulting in the downregulation of GPR17 transcription in SK-N-SH cells via PI3K/Akt/FoxO1 signaling (Figure 6).

## 3. Discussion

In the past decade, a large amount of evidence has accumulated to support a key role for GPR17 in neurodegeneration [23]. Bonfanti et al. [24] found that pathologic upregulation of GPR17 in the spinal cord of pre-symptomatic mice with amyotrophic lateral sclerosis contributes to oligodendrocyte dysfunction. Antagonism of GPR17 was found to successfully promote remyelination and cellular repair [25]. Zhao et al. [15] assessed GPR17 in oxygen–glucose deprivation-/recovery-induced ischemia-like injury in neuro-glial mixed cultures of cortical cells and found that GPR17 mediates ischemia-like neuronal injury and microglial activation. Zhou et al. [26] found that combination therapy with hyperbaric oxygen and erythropoietin downregulates GPR17 expression and upregulates the expression of myelin basic protein, thus resulting in an attenuation of demyelination and an inhibition of neuronal apoptosis of the spinal cord. These studies all underscore that the downregulation of GPR17 expression is vital to neuronal recovery.

Due to their rapid proliferation, SK-N-SH human neuroblastoma cells are commonly used to study neuronal damage and neurodegenerative diseases [27,28,29]. To mimic the nerve injury under AIS, SK-N-SH cells were starved for 24 h. IGF-1 promotes nerve cell repair, regulates neuroinflammation, and exerts a direct neuroprotective effect [30,31]. Here, we found that GPR17 expressed on SK-N-SH cells is downregulated by IGF-1, which is also associated with FoxO1 downregulation. Interestingly, both a FoxO1 inhibitor and a GPR17 antagonist were found to promote SK-N-SH cell viability and proliferation in a manner similar to IGF-1. To further explore the relationship between FoxO1 and GPR17, we packaged lentivirus-mediated shRNA to reduce FoxO1 expression and used adenoviral packaging to promote FoxO1 expression. Silencing of FoxO1 in SK-H-SH cells was found to downregulate GPR17 mRNA expression, while the overexpression of FoxO1 was found to be associated with an upregulation of GPR17 mRNA expression. Results from WB analysis did not reveal GPR17 to be significantly regulated by FoxO1. Here, the negligible effect of FoxO1 on GPR17 protein expression was likely due to the lack of good anti-GPR17 antibodies. We initially intended to use both the abs112096 and orb385461 anti-GPR17 antibodies in this study; however, the results were not ideal. After repeated testing, only abs112096 was selected for experimental use. Zappelli et al. [32] used a home-made rabbit anti-GPR17 antibody for a co-immunoprecipitation assay and reported that the molecular weight of GPR17 was between 37 and 50 KDa. Results from Zhao et al. [16] indicated that the molecular weight of GPR17 is 41 KDa. However, our results indicated that the molecular weight of GPR17 is between 35 and 40 KDa, which is lower than the reported value. The molecular weight change of GPR17 protein in our study might be due to receptor post-translational modification, such as glycosylation. The discrepancy between mRNA and protein results may be due to GPR17 protein turnover [33]. However, to the best of our knowledge, there is no literature evidence to support the GPR17 protein turnover. At present, this is stated only as a hypothesis. Further study is needed to verify whether the discrepancy between mRNA and protein results is caused by protein turnover. Furthermore, based on our qPCR and ChIP findings, the GPR17 protein expression is also most likely regulated by FoxO1 directly.

The PI3K/Akt signaling cascade is one of the major signaling pathways downstream from the activation of IGF-1 tyrosine kinase activity [4]. The PI3K/Akt/Foxo1 pathway is involved in the neuroinflammation caused by a stroke [34,35]. However, the relationship between IGF-1 and FoxO1 in neuronal cells, such as SK-H-SH cells, remains unclear. To investigate whether IGF-1 downregulates the FoxO1/GPR17 axis via PI3K/Akt signaling in these cells, LY294002 was used to block PI3K/Akt signaling. The suppressive effect of IGF-1 was attenuated after the inhibition of PI3K/Akt signaling by LY294002. Interestingly, LY294002 could also downregulate the expression of GPR17. As GPR17 could also be regulated by other factors, such as ETS1, whether the PI3K inhibitor LY294002 can regulate GPR17 expression through other signaling pathways remains to be further explored [36]. In addition, our ChIP assay results further showed that IGF-1 significantly reduces the binding of FoxO1 to GPR17 promoter regions in SK-N-SH cells. These results demonstrate that IGF-1 downregulates GPR17 expression via PI3K/Akt/FoxO1 signaling, resulting in SK-N-SH cell proliferation.

Antithrombotic therapy is recommended for the prevention of a secondary stroke in the management of most patients without contraindications [37]. Cangrelor, a rapidly acting P2Y12 receptor antagonist mainly used as an antiplatelet agent, was also reported to act as a non-specific GPR17 antagonist [38,39]. Cangrelor has been reported to ameliorate pulmonary injury [40], alleviate pulmonary fibrosis [41], and regulate food intake [20]. Here, we found that the antagonism of GPR17 by cangrelor promotes the viability and proliferation of SK-N-SH cells. As such, IGF-1 enhancement or GPR17 antagonism has potential as a combination therapy to promote neuroregeneration, particularly in the setting of AIS.

This study had several limitations. Most importantly, only in vitro effects were validated, without further in vivo confirmation. We did not perform any staining of GPR17 in normal cells or cells with starvation. It is not clear if the GPR17 is present only upon stimulation or also in control conditions, with or without starvation. Although two GPR17 antibodies from different suppliers were used, a WB analysis of GPR17 yielded inconsistent results. As the starvation of SK-H-SH cells for 24 h resulted in extensive cellular damage, the expression of glyceraldehyde-3-phosphate dehydrogenase (GAPDH) in WB results was found to vary. These limitations, however, do not affect the validity of any of the present findings.

## 4. Materials and Methods

### 4.1. Cell Culture and Treatment

Human neuroblastoma SK-N-SH cells (ATCC, USA) were cultured in Minimum Essential Medium with 10% fetal bovine serum and 100 μg/mL penicillin–streptomycin (Procell Life Science & Technology Co., Ltd., Wuhan, Hubei, China) at 37 °C with 5% CO2 in a humidified incubator. The cells were placed overnight in serum-free Minimum Essential Medium after medium replacement and treated with different concentrations (0, 25, 50, 100, and 200 ng/mL) of human IGF-1 (Bio-Techne, Minneapolis, MN, USA) at different time points (0, 1, 3, 6, 12, and 24 h), with or without a PI3K inhibitor (LY294002, 50 μmol/L, Cell Signaling #9901; Cell Signaling Technology subsidiary in China, Pudong, Shanghai, China). The cells were subsequently collected for further study.

### 4.2. Measurement of Cell Viability

The CellTiter-Lumi™ Luminescent Cell Viability Assay Kit (Beyotime, Songjiang, Shanghai, China) was used to quantify ATP content. Briefly, after SK-N-SH cells were starved for 24 h, they were treated with a FoxO1 inhibitor (AS1842856) at concentrations of 0, 50, and 100 nmol/L (MedChemExpress #HY-100596; MedChemexpress CO., Ltd., Pudong, Shanghai, China); a GPR17 antagonist (cangrelor) at concentrations of 0, 5, and 10 μmol/L (Sigma#SML2004; Sigma–Aldrich, St. Louis, MO, USA); and IGF-1 with or without a PI3K inhibitor (LY294002) at a concentration of 50 μmol/L. The cells were incubated with the provided reaction mixture for 10 min at room temperature. Absorbance was measured using a microtiter plate reader with a chemiluminescence function. The relative cell viability was determined after normalizing to untreated cells. Each treatment was carried out at least three times.

### 4.3. Cell Proliferation

Propagation of SK-N-SH cells was assessed using the 5-ethynyl-2-deoxyuridine (EdU) assay. SK-N-SH cells (3 × 105 per well) were seeded in a 24-well plate with tissue-culture-treated cell slides (diameter 14 mm) and then processed as described in Methodology 2.2. Before measurement, 10 mM of EdU was added to the medium and cells were incubated for 4 h. The cells were subsequently fixed with 4% paraformaldehyde and stained using the BeyoClicktEdU Cell Proliferation Kit with Alexa Fluor 488 and Hoechst 33342 (nuclear stain). Procedures were followed according to manufacturer instructions, and results were quantified using ImageJ software.

### 4.4. Construction of Lentivirus-Mediated Short Hairpin RNA (shRNA)

Lentivirus-mediated short hairpin RNA (shRNA) was constructed to mediate silencing of *FOXO1* in SK-N-SH cells. The PLKO.1 plasmids containing shRNAs targeting human *FOXO1* were purchased from GeneChem (Pudong, Shanghai, China). Lentivirus encoding relevant shRNA was generated over 72 h in HEK293T cells using the packaging vectors PMD2.0 and PSPAX2. The supernatant, containing recombinant lentivirus, was collected and purified. All shRNA sequences used were available upon request. Transfection was performed using the supernatant, containing recombinant lentivirus, according to manufacturer instructions. Stable clones expressing shRNA were then selected using puromycin (1 μg/mL). The levels of FoxO1 expression in selected clones were determined using Western blot (WB) and quantitative reverse-transcription PCR (RT-qPCR).

### 4.5. Construction of Adenoviral Vectors

Adenoviral vectors were used to induce the overexpression of FoxO1. Using the shuttle vector pAdTrack FKHR AAA (Addgene, 9037), Ad-FoxO1-AAA (3 Akt phosphorylation sites altered from serine/threonine to alanine) was generated. The resultant plasmid Adtrack-CMV-FKHR, linearized by Pme I, was co-transfected with pAdEasy-1 (adenoviral backbone plasmid) for a homologous recombination in E. coli BJ5183 cells. Positive recombinants were digested with Pac I and then transfected into HEK293A cells for adenoviral packaging. The pAdtrack-CMV vector was used as a control vector during viral packaging. Cells and culture media were harvested until the cytopathic effect became apparent. Three cycles of the freeze-and-thaw method were performed to release the adenovirus into the culture medium. After centrifugation, the supernatant, containing the virus, was subsequently precipitated with 5 × PEG buffer. The virus was then suspended in PBS containing 4% sucrose and aliquoted to vials before storage at −80 °C. SK-N-SH cells were infected with recombinant adenovirus expressing either a control (pAdtrack-CMV) or a test (pAdTrack FKHR AAA) virus for 24 h. After transfection, FoxO1 expression levels were determined by WB and RT-qPCR.

### 4.6. RT-qPCR

RT-qPCR was used to quantify *GPR17* and *FOXO1* expression in SK-N-SH cells. After treatment, the cells were harvested and total RNA extracted using a trizol reagent (Invitrogen, Carlsbad, CA, USA). Total RNA was subsequently reverse-transcribed into cDNA using an RT Reagent kit (Takara Bio, Dalian, Liaoning, China). Real-time polymerase chain reaction (qPCR) amplification was performed using a Bio-rad CFX Connect fluorescent QPCR system according to manufacturer instructions. The relative expression of target genes was normalized to β-actin. Mean cycle threshold values of each group obtained from triplicate measurement were recorded and determined using the 2-ΔΔCt formula. The primer sequences are shown in Table 1.

### 4.7. Western Blot Analysis

WB analysis was used to quantify the expression of FoxO1, p-FoxO1, Akt, p-Akt, GPR17, and GAPDH active in signaling. After different treatments, total protein from SK-N-SH cells was extracted at 4 °C and the protein concentration quantified using a BCA protein assay kit. For each sample, 40 μg of protein was loaded. The following primary antibodies were used: anti-FoxO1 antibody (1:1000, Cell Signaling #2880), anti-p-FoxO1 Ser256 antibody (1:1000, Cell Signaling #9461), anti-Akt antibody (1:1000, Cell Signaling #4691), anti-p-Akt Ser473 antibody (1:1000, Cell Signaling #9271), anti-GPR17 antibody (1:500, Absin Bioscience Inc. #abs112096, Shanghai, China), and anti-GAPDH antibody (1:5000, Abcam #181602). The second antibody used was Goat anti-Rabbit IgG H&L (IRDye*^®^* 800 CW; 1:10,000, Abcam #216773). An Odyssey fluorescent scanner (LI-COR, Bioscience, Lincoln, NE, USA) was used to detect protein bands. The relative fold-change of protein expression levels was calculated by normalization to GAPDH levels using Quantity One software (Bio-Rad Laboratories, Inc., Hercules, CA, USA).

### 4.8. Immunofluorescence Cell Staining

SK-N-SH cells were seeded in a 24-well plate pre-laid with TC-treated cell slides (diameter: 14 mm). After cultivation in a serum-free medium, the cells were treated with 100 ng/mL IGF-1 in the presence or absence of 50 μM of LY294001 for 24 h and fixed with 4% paraformaldehyde. The plasma membrane was permeabilized with 0.1% Triton X-100 for 30 min, and the slides were blocked with 2% BSA in 0.01 M PBS (pH 7.3) for 30 min. After overnight incubation at 4 °C with the anti-FoxO1 antibody (rabbit, 1:100; Cell Signaling Technology subsidiary in China, Pudong, Shanghai, China), the cells were stained using a single-color (Cy3) fluorescence kit (Shanghai Recordbio Biological Technology, Shanghai, China) for 10–15 min at room temperature. After extensive washing, the nuclei were stained with DAPI and scanned using a Leica SP5 confocal laser scanning microscope (Leica, Wetzlar, Germany) equipped with a ×63 objective. Two channels were used to sequentially acquire images: Cy3laser (520 nm; red for FoxO1) and DAPI (405 nm; blue for nuclei). Images were captured separately and then merged. The cells exhibiting co-localization of signals generated from both channels were identified as expressing FoxO1 in the nuclei.

### 4.9. Chromatin Immunoprecipitation qPCR

Chromatin immunoprecipitation (ChIP) assays were conducted using the SimpleChIP*^®^* Enzymatic Chromatin IP kit (Agarose Beads; Cell Signaling Technology subsidiary in China, Pudong, Shanghai, China) according to manufacturer instructions. De-cross-linked DNA samples were subjected to PCR amplification using forward (5′-CACGGAGCCTAAGTTCTGG-3′) and reverse (5′-TGAGGCTCAGGCAAATGAA-3′) primers targeting the GPR17 promoter. Precipitated DNA fragments were analyzed using qPCR. The signal obtained from each immunoprecipitation was expressed as a percentage of the total chromatin input. The following formula was used: input percentage (signal relative to input) = 2% × 2^ (C(T) 2% input sample C(T) IP sample).

### 4.10. Statistical Analyses

Statistical analyses were performed using GraphPad Prism software 6.0 (GraphPad Software Inc., San Diego, CA, USA). Data were presented as the mean ± SD. Data comparisons among groups were performed using one-way ANOVA and the independent sample *t*-test. *p* < 0.05 was considered statistically significant.

## 5. Conclusions

In summary, the present study confirmed that IGF-1 promotes SK-N-SH cell proliferation associated with FoxO1 and GPR17 downregulation. We demonstrated that IGF-1 acts in P13K/Akt signaling to enhance SK-N-SH cellular viability and proliferation via the promotion of FoxO1 nuclear export and reduction in the binding of FoxO1 to the GPR17 promoter. These findings suggest that the enhancement of IGF-1 signaling and the antagonism of GPR17 serve as a potential therapeutic strategy in the management of neuronal injury conditions, such as acute ischemic stroke.

## Figures and Tables

**Figure 1 ijms-23-01513-f001:**
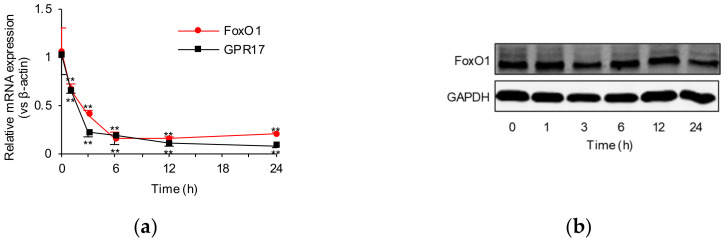
IGF-1 regulates the expression of FoxO1 and GPR17 in SK-N-SH cells. (**a**) *FOXO1* and *GPR17* mRNA expression induced by 100 ng/mL of IGF-1 within 24 h (1, 3, 6, 12, and 24 h); (**b**) Western blot analysis of FoxO1 expression induced by 100 ng/mL of IGF-1 within 24 h; (**c**) quantification of FoxO1 protein levels induced by 100 ng/mL of IGF-1 within 24 h; (**d**) influence of IGF-1 at different concentrations (0, 25, 50, 100, and 200 ng/mL) on *FOXO1* and *GPR17* mRNA expression at 24 h; (**e**) Western blot analysis of FoxO1 protein expression induced by different concentrations of IGF-1 (0, 25, 50, 100, and 200 ng/mL) at 24 h; (**f**) quantification of FoxO1 protein expression induced by different concentrations of IGF-1 (0, 25, 50, 100, and 200 ng/mL) at 24 h. Results were obtained from three separate experiments. ** *p* < 0.01 vs. 0 h or 0 ng/mL in the IGF-1 group.

**Figure 2 ijms-23-01513-f002:**
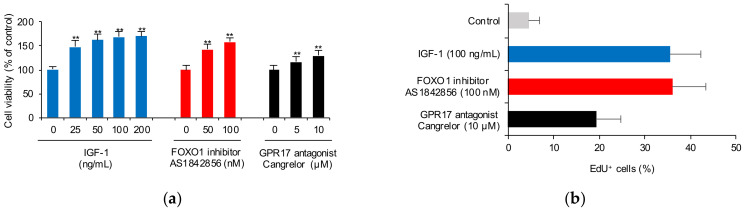
SK-N-SH cell viability and proliferation after treatment with IGF-1, a FoxO1 inhibitor and a GPR17 antagonist. (**a**) Viability of cells after treatment with IGF-1 (0, 25, 50, 100, and 200 ng/mL), a FoxO1 inhibitor (AS1842856; 0, 50, and 100 nM), and a GPR17 antagonist (cangrelor; 0, 5, and 10 μM); (**b**) cellular proliferation induced by 100 ng/mL of IGF-1, 100 nM of AS1842856, and 10 μM of cangrelor; (**c**) representative images of 5-ethynyl-2-deoxyuridine (EdU) staining in cells with or without treatment with 100 ng/mL of IGF-1, 100 nM of AS1842856, and 10 μM of cangrelor. (Scale bar: 20 μm) ** *p* < 0.01 vs. control.

**Figure 3 ijms-23-01513-f003:**
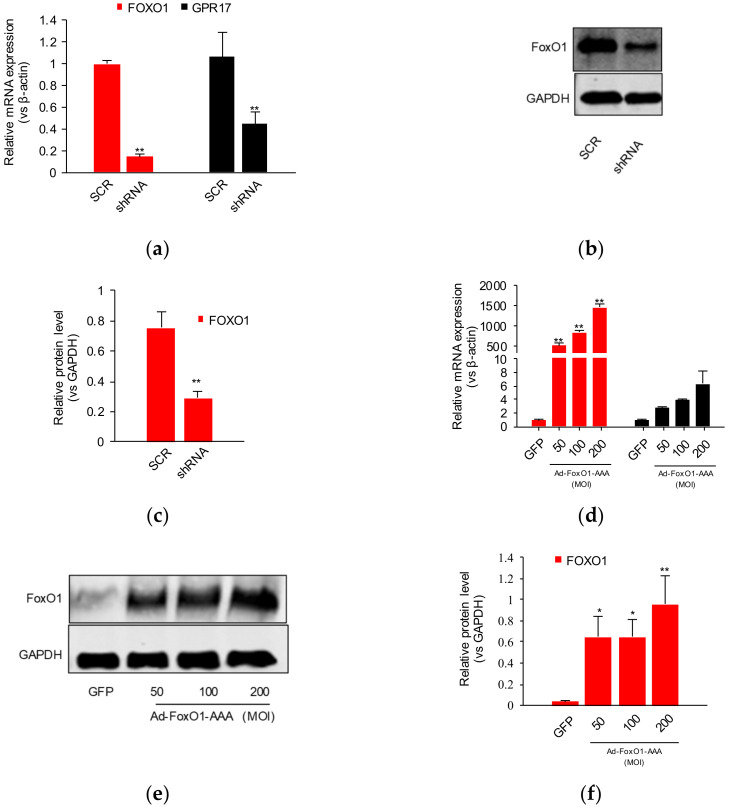
The influence of FoxO1 silencing and GPR17 overexpression on SK-N-SH cells. (**a**) The influence of FoxO1 silencing on *FOXO1* and *GPR17* mRNA expression; (**b**) Western blot analysis of FoxO1 protein expression after FoxO1 silencing; (**c**) quantification of FoxO1 protein levels after FoxO1 silencing; (**d**) the influence of FoxO1 overexpression on FoxO1 and GPR17 mRNA levels; (**e**) Western blot analysis of FoxO1 protein expression after FoxO1 overexpression; (**f**) quantification of FoxO1 protein levels after FoxO1 overexpression. Results were obtained from three separate experiments. * *p* < 0.05 vs. control or ** *p* < 0.01 vs. control (SCR, GFP).

**Figure 4 ijms-23-01513-f004:**
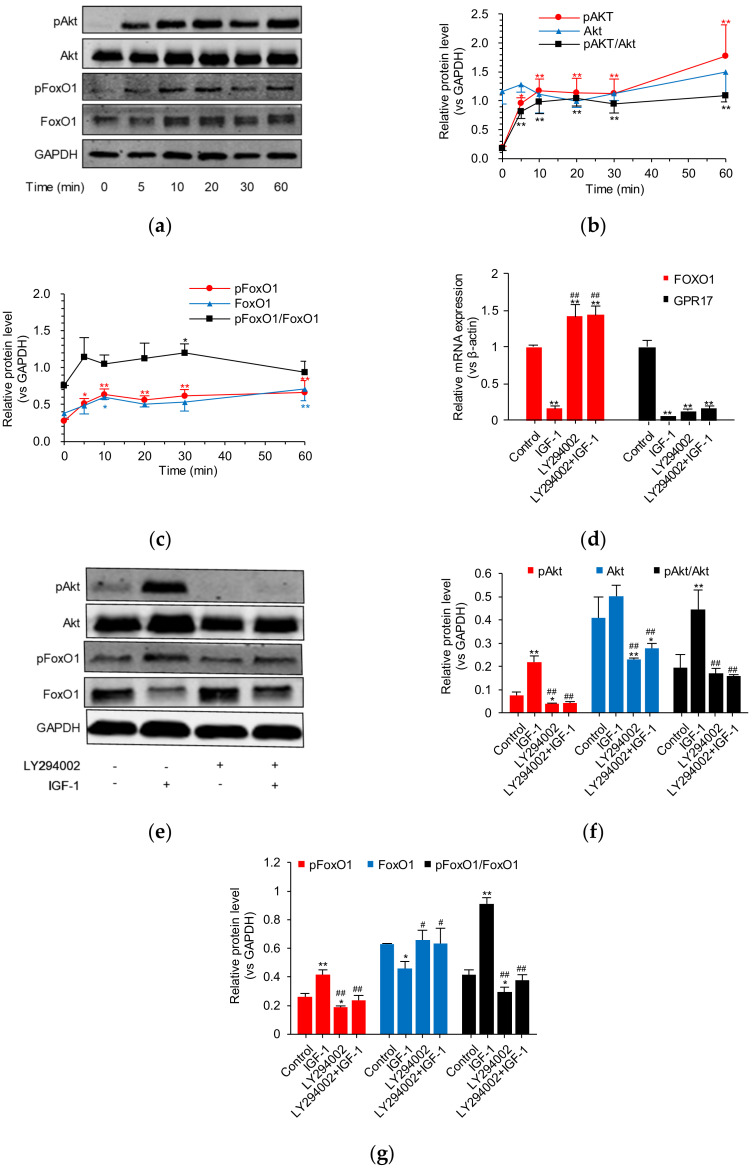
The influence of IGF-1 on PI3K/Akt/FoxO1-GPR17 signaling in SK-N-SH cells. (**a**) Western blot analysis of pAkt, Akt, pFoxO1, and FoxO1 protein expression in cells treated with 100 ng/mL of IGF-1 within 60 min; (**b**,**c**) quantification of pAkt, Akt, pFoxO1, and FoxO1 protein levels within 60 min; (**d**) levels of FoxO1 and GPR17 mRNA expressed in cells with or without IGF-1 and PI3K inhibitor (LY294002) treatment; (**e**) Western blot analysis of pAkt, Akt, pFoxO1, and FoxO1 expression in SK-N-SH cells with or without IGF-1 and PI3K inhibitor (LY294002) treatment. “+” means “with”, “-” means “without”; (**f**,**g**) quantification of pAkt, Akt, pFoxO1, and FoxO1 levels in cells with or without IGF-1 and PI3K inhibitor (LY294002) treatment. Results were obtained from three separate experiments. Data are presented as the mean ± SD. * *p* < 0.05 vs. 0 min or control; ** *p* < 0.01 vs. 0 min or control. ^#^ *p* < 0.05 vs. the IGF-1 group; ^##^ *p* < 0.01 vs. the IGF-1 group.

**Figure 5 ijms-23-01513-f005:**
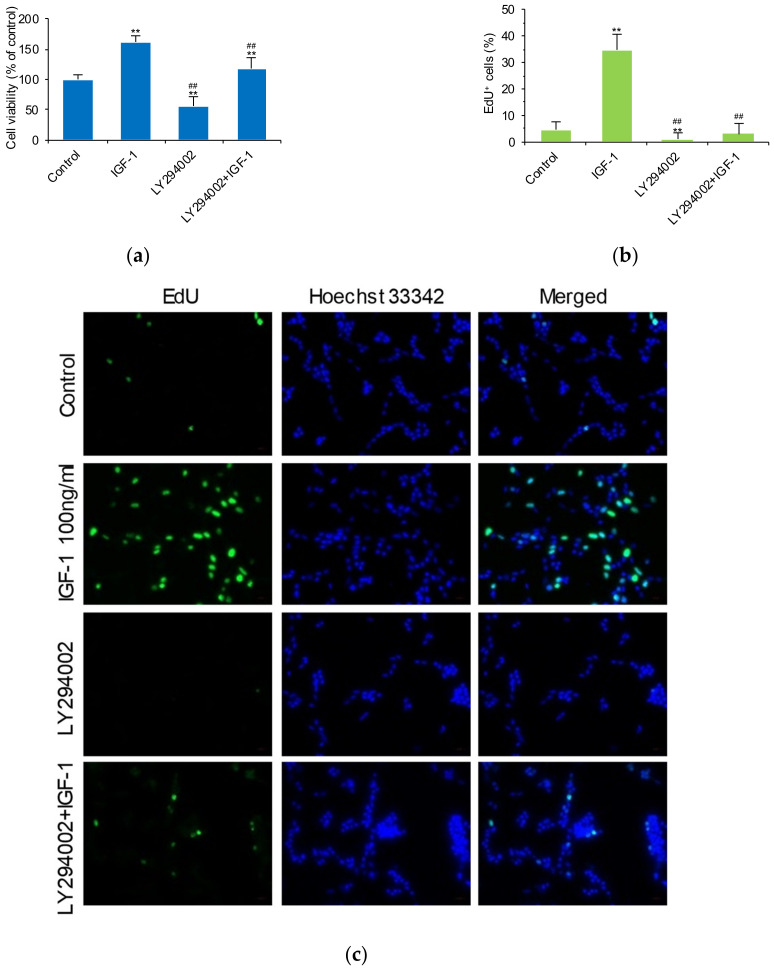
IGF-1 acts in the P13K/AKT pathway to promote SK-N-SH cellular proliferation and inhibit the nuclear translocation of FoxO1. (**a**) SK-N-SH cell viability with or without IGF-1 and P13K inhibitor (LY294002) treatment; (**b**) Edu+ cell percentage as determined by 5-ethynyl-2-deoxyuridine (EdU) staining in cells with or without IGF-1 and P13K inhibitor (LY294002) treatment; (**c**) representative images of EdU staining of cells with or without treatment with 100 ng/mL of IGF-1 and a P13K inhibitor (LY294002). The nuclei were stained blue; the nuclei of cells possessing high levels of DNA replication (EdU+ cells) were simultaneously stained green. Scale bar: 20 μm; (**d**) representative images of FoxO1 staining of cells with or without IGF-1 and P13K inhibitor (LY294002) treatment. Scale bar: 50 or 20 μm; (**e**,**f**) chromatin immunoprecipitation assay results of *FOXO1* binding to the promoter regions of *GPR17*. Data are presented as a signal relative to the FoxO1 (C29H4) Rabbit mAb #2880/Normal Rabbit IgG #2729 input ratio. Normal rabbit IgG #2729 is an internal reference. The ratio is used to determine the binding rate of *FOXO1* to the promoter regions of *GPR17*. Cells with or without (control) treatment with 100 ng/mL of IGF-1 after starvation for 24 h (**e**); normal cells with or without *FOXO1* adenoviral vector treatment (green fluorescent protein, GFP; control) (**f**). Results were obtained from three separate experiments. Data are presented as the mean ± SD. * *p* < 0.05 vs. control; ** *p* < 0.01 vs. control; ^##^ *p* < 0.01 vs. the IGF-1 group.

**Figure 6 ijms-23-01513-f006:**
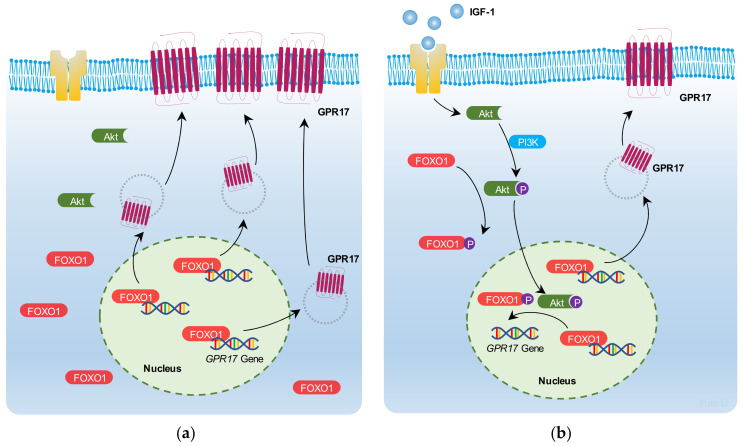
IGF-1/PI3K/Akt/FoxO1/GPR17 signaling. The expression of the GPR17 receptor on the cell membrane of SK-N-SH cells is regulated by IGF-1/PI3K/Akt/FoxO1 signaling. (**a**) In starved SK-N-SH cells, the expression of GPR17 receptor on the cell membrane is increased. This may be because the binding of *FOXO1* to the promoter regions of *GPR17* enhances GPR17 expression; (**b**) IGF-1 could promote the proliferation of SK-N-SH cells. This may be due to the action of IGF-1 receptor by IGF-1, followed by an increase in Akt phosphorylation (pAkt). pAkt promotes *FOXO1* phosphorylation and reduces *FOXO1* binding to the promoter regions of *GPR17*, thus resulting in the downregulation of GPR17 expression in starved SK-N-SH cells.

**Table 1 ijms-23-01513-t001:** Primer sequences of human *FOXO1*, *GPR17*, and *β-actin* genes.

Genes	Forward Primer (5′-3′)	Reverse Primer (5′-3′)
*FOXO1*	TGTCCTACGCCGACCTCATCAC	GCACGCTCTTGACCATCCACTC
*GPR17*	GTTGGCAATACCCTGGCTCTGTG	GGACCAGCACGCACGACAAG
*β-actin*	TGGACTTCGAGCAAGAGATG	GAAGGAAGGCTGGAAGAGTG

## Data Availability

The data presented in this study are available in the Appendix A.

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
