# Peer review of "Insulin-like Growth Factor 1 Promotes Cell Proliferation by Downregulation of G-Protein-Coupled Receptor 17 Expression via PI3K/Akt/FoxO1 Signaling in SK-N-SH Cells"

_ijms, 2022, doi:10.3390/ijms23031513_

Round 1

Reviewer 1 Report

Congrats for your research!

 I recommend the following modifications:

  • Please use the full name and the same abbreviations , for example line 20 PI3K, line 25 for AkT like in the title, line 50, 146, 148-1508, 239,234,23
  • IGF 1 is already abbreviated at line 43 so, it is not necessary once again at line 64
  • Idem GPR17, line 42 and 73
  • Line 90 please explain the abbreviation WB
  • Line 161 pAkt and sometimes PAkT
  • Figure 6 –the legend is too short, more details or explanations
  • Line 210-please explain what kind of study performed Zhao B et al. [15], what kind of cell line they used
  • Line 257- please explain GADPH or GAPDH in line 340
  • Please improve the conclusions

Author Response

We thank the reviewer for the positive comments about our study. We thank the reviewer for these valuable advises. We revised our manuscript according to the reviewer’s suggestion carefully.

Comment 1:  Please use the full name and the same abbreviations , for example line 20 PI3K, line 25 for AkT like in the title, line 50, 146, 148-1508, 239,234,23

Response:  We thank the reviewer’s valuable advises. We revised the full name and used the same abbreviations in the manuscript.

Comment 2: IGF 1 is already abbreviated at line 43 so, it is not necessary once again at line 64

Response:  We thank the reviewer’s valuable advises. We used its abbreviated form in the manuscript.

Comment 3: Idem GPR17, line 42 and 73.

Response:  We thank the reviewer’s valuable advises. We used its abbreviated form in the manuscript.

Comment 4: Line 90 please explain the abbreviation WB.

Response:  We thank the reviewer for pointing out it. WB was the abbreviation of Western blot. We added the full name in the manuscript.

Comment 5: Line 161 pAkt and sometimes PAkT.

Response:  We thank the reviewer for pointing out it. We uniformly modify it as pAkt in the article.

Comment 6: Figure 6 –the legend is too short, more details or explanations

Response:  We thank the reviewer for pointing out it. We improved the legend of Figure 6 as follows:

Figure 6. IGF-1/PI3K/Akt/FOXO1/GPR17 signaling.

The expression of GPR17 receptor on the cell membrane of SK-N-SH cells is regulated by IGF-1/PI3K/Akt/FOXO1 signaling. (a) In starved SK-N-SH cells, the expression of GPR17 receptor on the cell membrane is increased. This may be due to the binding of FOXO1 to the promoter regions of GPR17 enhances GPR17 expression; (b) IGF-1 could promote the proliferation of SK-N-SH cells. This may be due to the action of IGF-1 receptor by IGF-1, following by an increase in Akt phosphorylation (pAkt). pAkt promotes FOXO1 phosphorylation and reduces FOXO1 binding to the promoter regions of GPR17, thus resulting in the downregulation of GPR17 expression in starved SK-N-SH cells.

Comment 7: Line 210-please explain what kind of study performed Zhao B et al. [15], what kind of cell line they used

Response:  We thank the reviewer for pointing out it. The study from Zhao B et al. [15] assessed the potential mechanism of GPR17 in oxygen-glucose deprivation/recovery-induced ischemia-like injury in vitro. We revised it in the manuscript as follows:

Zhao B et al. [15] assessed GPR17 in oxygen-glucose deprivation/recovery-induced ischemia-like injury in neuro-glial mixed cultures of cortical cells and found that GPR17 mediates ischemia-like neuronal injury and microglial activation.

Comment 8: Line 257- please explain GADPH or GAPDH in line 340

Response:  We thank the reviewer for pointing out it. We added the full name of GAPDH in the manuscript. GADPH was a spelling mistake. We have corrected it in the manuscript.

Comment 9: Please improve the conclusions

Response:  We thank the reviewer for pointing out it. We improved the conclusion in the manuscript as follows:

In summary, the present study confirmed that IGF-1 promotes SK-N-SH cell proliferation associated with FoxO1 and GPR17 downregulation. We demonstrate that IGF-1 acts in P13K/Akt signaling to enhance SK-N-SH cellular viability and proliferation via the promotion of FoxO1 nuclear export and reduction in the binding of FoxO1 to the GPR17 promoter. These findings suggest that enhancement of IGF-1 signaling and antagonism of GPR17 serves as a potential therapeutic strategy in the management of neuronal injury condition such as acute ischemic stroke.

Reviewer 2 Report

In the manuscript by Lin et al., the authors investigate the effect of IGF-1 on the expression of GPR17, a membrane receptor previously described a sensor of damage in several disease models including brain ischemia, in the neuroblastoma cell line SK-N-SH. Using some basic pharmacological and biotechnological approaches, the authors identify a potential axis that links IGF-1 to GPR17 through PI3K-AKT and that influence neuroblastoma cell proliferation. Due to the already described involvement of GPR17 in proliferation/differentiation pathways both in physiological and pathological conditions, the authors reason that the stimulation of this axis may prevent GPR17 pathological overexpression in disease.

The manuscript is interesting, well written and the rationale is clearly explained, but some results are contradictory, and some important points are missing. Please, find below my major comments:

  1. 2.1 and Fig.1: the authors describe for the first time the expression of GPR17 in SK-N-SH. It is not clear if the receptor is present only upon stimulation or also in control conditions with or without starvation. This starting point would help to contextualize the receptor. Some stainings would also help the reader to understand if the results on GPR17 protein are trustworthy. Indeed, the authors mention several times in the text that the tested antibodies did not work properly.
  2. Lines 159-160: the conclusion of this paragraph is not clear. The authors claim that IGF-1 downregulates GPR17 expression through PI3K-AKT-FOXO1, but no reduction in GPR17 was found in WB. How do the authors reconcile these data? As mentioned by the authors in the discussion, the lack of a good antibody is a limit that reduce the reliability of the results. Is the WB band compatible with the molecular weight of the protein? Otherwise, the results are not trustworthy and should be removed. GPR17 is not a common marker, the mw of the protein should be reported in the figures. Any means to show GPR17 protein expression would strengthen their results.
  3. The authors should also consider the protein turnover: is GPR17 turnover sufficiently fast to claim that the reduction of the transcript is relevant for cell functions? They should briefly comment this issue in the discussion.
  4. Figure 4d: The authors show that GPR17 mRNA is dramatically downregulated in all the conditions. How do they explain this apparent contradiction? A brief explanation should be included.
  5. Figure 5e,f: The Y axis is not explained. Text and caption do not help. If the black bars in f are GPR17 with and without FOXO1, the results do not confirm FOXO1 binding. The authors should clarify. The binding between FOXO1 and GPR17 promoter has been demonstrated in very different experimental settings. Not necessarily this mechanism is present (or quantitatively relevant) in cell lines. Do the authors exclude that the effect on GPR17 is indirect?

Minor comments:

  1. The first paragraph of the Introduction is misleading and should be removed since the manuscript is not about acute ischemic stroke.
  2. In Fig 4f, captions are missing. This is confusing since not all the red bars represent the same actor in the figure.
  3. I suggest labelling figure panels at the upper left side. Bottom center (as they appear now) may be misleading for some readers.

Author Response

We thank the reviewer for the positive comments about our study. We thank the reviewer for these valuable advises. We revised our manuscript according to the reviewer’s suggestion carefully.

Comment 1: 2.1 and Fig.1: the authors describe for the first time the expression of GPR17 in SK-N-SH. It is not clear if the receptor is present only upon stimulation or also in control conditions with or without starvation. This starting point would help to contextualize the receptor. Some stainings would also help the reader to understand if the results on GPR17 protein are trustworthy. Indeed, the authors mention several times in the text that the tested antibodies did not work properly.

Response:  We thank the reviewer for pointing it. In our study, we did not analyze whether GPR17 receptor was present only upon SK-N-SH cell starved treated or also in control conditions. So, we did not perform any staining of GPR17 receptor in normal cells or starved treated cells. We add these to the limitations in the discussion section as follows:

This study had several limitations. Most importantly, only in vitro effects were validated without further in vivo confirmation. We did not perform any staining of GPR17 receptor in normal cells or cells with starvation. It is not clear if the GPR17 receptor is pre-sent only upon stimulation or also in control conditions with or without starvation. Alt-hough two kinds of GPR17 antibodies were utilized, WB of GPR17 yielded inconsistent results. As starvation of SK-H-SH cells for 24h resulted in extensive cellular damage, expression of glyceraldehyde-3-phosphate dehydrogenase (GAPDH) in WB results was found to vary. These limitations, however, do not affect the validity of any of the afore-mentioned findings.

Comment 2: Lines 159-160: the conclusion of this paragraph is not clear. The authors claim that IGF-1 downregulates GPR17 expression through PI3K-AKT-FOXO1, but no reduction in GPR17 was found in WB. How do the authors reconcile these data? As mentioned by the authors in the discussion, the lack of a good antibody is a limit that reduce the reliability of the results. Is the WB band compatible with the molecular weight of the protein? Otherwise, the results are not trustworthy and should be removed. GPR17 is not a common marker, the mw of the protein should be reported in the figures. Any means to show GPR17 protein expression would strengthen their results.

Response:  We thank the reviewer for pointing out it. GPR17 is not a common marker, the lack of a good antibody is a limit that reduce the reliability of the WB results. We have attached the original pictures of all WB results including GPR17 WB results for reviewers and readers. The mw of GPR17 and the marker were reported in the supplemental materials. We also added some sentences about it in the Discussion as follows:

Zappelli E et al.[32] used a home-made rabbit anti-GPR17 antibody for co-immunoprecipitation assay and reported that the molecular weight of GPR17 was between 37 and 50 KDa. Results from Zhao B et al.[16] indicated that the molecular weight of GPR17 was 41 KDa. However, our results indicated that the molecular weight of GPR17 was between 35 and 40 KDa, which was lower than the reported value. The molecular weight change of GPR17 protein in our study may be associated with GPR17 protein turnover[33]. However, to the best of our knowledge, there is no evidence to support the GPR17 protein turnover hypothesis.

Comment 3: The authors should also consider the protein turnover: is GPR17 turnover sufficiently fast to claim that the reduction of the transcript is relevant for cell functions? They should briefly comment this issue in the discussion.

Response:  We thank the reviewer for pointing out it. We added a briefly comment about issue in the discussion as follows:

The molecular weight change of GPR17 protein in our study may be associated with GPR17 protein turnover[33]. However, to the best of our knowledge, there is no evidence to support the GPR17 protein turnover hypothesis.

Comment 4: Figure 4d: The authors show that GPR17 mRNA is dramatically downregulated in all the conditions. How do they explain this apparent contradiction? A brief explanation should be included.

Response:  We thank the reviewer for pointing out it. We added a brief explanation in the Discussion as follows: Interestingly, LY294002 could also downregulate the expression of GPR17. As GPR17 could also be regulated by other factors such as ETS1, whether the PI3K inhibi-tor LY294002 regulates the GPR17 expression through other signaling pathway still needs to be further explored.

Comment 5: Figure 5e,f: The Y axis is not explained. Text and caption do not help. If the black bars in f are GPR17 with and without FOXO1, the results do not confirm FOXO1 binding. The authors should clarify. The binding between FOXO1 and GPR17 promoter has been demonstrated in very different experimental settings. Not necessarily this mechanism is present (or quantitatively relevant) in cell lines. Do the authors exclude that the effect on GPR17 is indirect?

Response:  We thank the reviewer for pointing out it. We added the Y axis of Figure 5e and 5f in the manuscript.

Minor comments:

Comment 6:  The first paragraph of the Introduction is misleading and should be removed since the manuscript is not about acute ischemic stroke.

Response:  We thank the reviewer for pointing out it. Acute ischemic stroke (AIS) is defined as sudden focal ischemia of brain tissue lasting for over 24 hours and resulting in neurologic dysfunction. That's why we treat SK-H-SH cell starvation for 24 hours. Therefore, we did not remove the first paragraph of the Introduction. We added an explanation to this in the Discussion section as follow:

To mimic the nerve injury under AIS, SK-N-SH cells were starved for 24 hours.

Comment 7: In Fig 4f, captions are missing. This is confusing since not all the red bars represent the same actor in the figure.

Response:  We thank the reviewer for pointing out it. We added the captions in the Figure 4f.

Comment 8: I suggest labelling figure panels at the upper left side. Bottom center (as they appear now) may be misleading for some readers.

Response:  We thank the reviewer for this suggestion. However, our manuscript was prepared based on the template of the journal. The template figure panels of this journal are bottom center. So, we did not change it.

Round 2

Reviewer 2 Report

Comment 1: 2.1 and Fig.1: the authors describe for the first time the expression of GPR17 in SK-N-SH. It is not clear if the receptor is present only upon stimulation or also in control conditions with or without starvation. This starting point would help to contextualize the receptor. Some stainings would also help the reader to understand if the results on GPR17 protein are trustworthy. Indeed, the authors mention several times in the text that the tested antibodies did not work properly.

Response: We thank the reviewer for pointing it. In our study, we did not analyze whether GPR17 receptor was present only upon SK-N-SH cell starved treated or also in control conditions. So, we did not perform any staining of GPR17 receptor in normal cells or starved treated cells. We add these to the limitations in the discussion section as follows:

This study had several limitations. Most importantly, only in vitro effects were validated without further in vivo confirmation. We did not perform any staining of GPR17 receptor in normal cells or cells with starvation. It is not clear if the GPR17 receptor is pre-sent only upon stimulation or also in control conditions with or without starvation. Alt-hough two kinds of GPR17 antibodies were utilized, WB of GPR17 yielded inconsistent results. As starvation of SK-H-SH cells for 24h resulted in extensive cellular damage, expression of glyceraldehyde-3-phosphate dehydrogenase (GAPDH) in WB results was found to vary. These limitations, however, do not affect the validity of any of the afore-mentioned findings.

If WB of GPR17 yielded inconsistent results, they should not be presented. In the manuscript some figures based their results on them. These results cannot be presented as regular figures. For the sake of the readers, the data should be presented as supplementary.

Molecular weight should be shown for any markers (including FOXO1, AKT, etc).

Comment 2: Lines 159-160: the conclusion of this paragraph is not clear. The authors claim that IGF-1 downregulates GPR17 expression through PI3K-AKT-FOXO1, but no reduction in GPR17 was found in WB. How do the authors reconcile these data? As mentioned by the authors in the discussion, the lack of a good antibody is a limit that reduce the reliability of the results. Is the WB band compatible with the molecular weight of the protein? Otherwise, the results are not trustworthy and should be removed. GPR17 is not a common marker, the mw of the protein should be reported in the figures. Any means to show GPR17 protein expression would strengthen their results.

Response: We thank the reviewer for pointing out it. GPR17 is not a common marker, the lack of a good antibody is a limit that reduce the reliability of the WB results. We have attached the original pictures of all WB results including GPR17 WB results for reviewers and readers. The mw of GPR17 and the marker were reported in the supplemental materials. We also added some sentences about it in the Discussion as follows:

Zappelli E et al.[32] used a home-made rabbit anti-GPR17 antibody for co-immunoprecipitation assay and reported that the molecular weight of GPR17 was between 37 and 50 KDa. Results from Zhao B et al.[16] indicated that the molecular weight of GPR17 was 41 KDa. However, our results indicated that the molecular weight of GPR17 was between 35 and 40 KDa, which was lower than the reported value. The molecular weight change of GPR17 protein in our study may be associated with GPR17 protein turnover[33]. However, to the best of our knowledge, there is no evidence to support the GPR17 protein turnover hypothesis.

It is unlikely that the change in molecular weight is related to protein turnover. It might be due to receptor post-translational modification (e.g., glycosylation). I recommend that the authors do not mention this sentence in the text. They should clearly label the molecular weight in the figures and describe the discrepancy as a different glycosylation state. This discrepancy does not necessarily relate to the reliability of their data since different cells may introduce different glycosylation.

Comment 3: The authors should also consider the protein turnover: is GPR17 turnover sufficiently fast to claim that the reduction of the transcript is relevant for cell functions? They should briefly comment this issue in the discussion.

Response: We thank the reviewer for pointing out it. We added a briefly comment about issue in the discussion as follows:

The molecular weight change of GPR17 protein in our study may be associated with GPR17 protein turnover[33]. However, to the best of our knowledge, there is no evidence to support the GPR17 protein turnover hypothesis.

See my previous comment. To me, this sentence makes no sense. If the authors do not trust their own data, because of the lack of reproducibility they should not comment on that. Otherwise, they should not exclude that discrepancy between mRNA and protein results are due to protein turnover. If there is no literature evidence about this hypothesis, it should be stated only as a hypothesis.

Comment 5: Figure 5e,f: The Y axis is not explained. Text and caption do not help. If the black bars in f are GPR17 with and without FOXO1, the results do not confirm FOXO1 binding. The authors should clarify. The binding between FOXO1 and GPR17 promoter has been demonstrated in very different experimental settings. Not necessarily this mechanism is present (or quantitatively relevant) in cell lines. Do the authors exclude that the effect on GPR17 is indirect?

Response: We thank the reviewer for pointing out it. We added the Y axis of Figure 5e and 5f in the manuscript.

The overall meaning of the “RATIO” should be explained. Why have the authors normalized vs. Normal Rabbit IgG? It may not be clear for those who are not familiar with the ChIP. Then, I suggest to simplify the label on the Y-axis, removing the product codes.

Minor comments:

Comment 6: The first paragraph of the Introduction is misleading and should be removed since the manuscript is not about acute ischemic stroke.

Response: We thank the reviewer for pointing out it. [...] Therefore, we did not remove the first paragraph of the Introduction. We added an explanation to this in the Discussion section as follow:

To mimic the nerve injury under AIS, SK-N-SH cells were starved for 24 hours.

I see. In this case, I suggest that the authors add a brief mention also at the end of the introduction, otherwise, that paragraph seems off-topic.

Author Response

Comment 1: 2.1 and Fig.1: the authors describe for the first time the expression of GPR17 in SK-N-SH. It is not clear if the receptor is present only upon stimulation or also in control conditions with or without starvation. This starting point would help to contextualize the receptor. Some stainings would also help the reader to understand if the results on GPR17 protein are trustworthy. Indeed, the authors mention several times in the text that the tested antibodies did not work properly.

Response: We thank the reviewer for pointing it. In our study, we did not analyze whether GPR17 receptor was present only upon SK-N-SH cell starved treated or also in control conditions. So, we did not perform any staining of GPR17 receptor in normal cells or starved treated cells. We add these to the limitations in the discussion section as follows:

This study had several limitations. Most importantly, only in vitro effects were validated without further in vivo confirmation. We did not perform any staining of GPR17 receptor in normal cells or cells with starvation. It is not clear if the GPR17 receptor is pre-sent only upon stimulation or also in control conditions with or without starvation. Alt-hough two kinds of GPR17 antibodies were utilized, WB of GPR17 yielded inconsistent results. As starvation of SK-H-SH cells for 24h resulted in extensive cellular damage, expression of glyceraldehyde-3-phosphate dehydrogenase (GAPDH) in WB results was found to vary. These limitations, however, do not affect the validity of any of the afore-mentioned findings.

New comment 1: If WB of GPR17 yielded inconsistent results, they should not be presented. In the manuscript some figures based their results on them. These results cannot be presented as regular figures. For the sake of the readers, the data should be presented as supplementary.

Molecular weight should be shown for any markers (including FOXO1, AKT, etc).

Response: We thank the reviewer for pointing out it. We moved all WB results of GPR17 to the supplementary file. All molecular weight for markers in our study were marked in supplementary file.

Changes in the text: See manuscript.

Comment 2: Lines 159-160: the conclusion of this paragraph is not clear. The authors claim that IGF-1 downregulates GPR17 expression through PI3K-AKT-FOXO1, but no reduction in GPR17 was found in WB. How do the authors reconcile these data? As mentioned by the authors in the discussion, the lack of a good antibody is a limit that reduce the reliability of the results. Is the WB band compatible with the molecular weight of the protein? Otherwise, the results are not trustworthy and should be removed. GPR17 is not a common marker, the mw of the protein should be reported in the figures. Any means to show GPR17 protein expression would strengthen their results.

Response: We thank the reviewer for pointing out it. GPR17 is not a common marker, the lack of a good antibody is a limit that reduce the reliability of the WB results. We have attached the original pictures of all WB results including GPR17 WB results for reviewers and readers. The mw of GPR17 and the marker were reported in the supplemental materials. We also added some sentences about it in the Discussion as follows:

Zappelli E et al.[32] used a home-made rabbit anti-GPR17 antibody for co-immunoprecipitation assay and reported that the molecular weight of GPR17 was between 37 and 50 KDa. Results from Zhao B et al.[16] indicated that the molecular weight of GPR17 was 41 KDa. However, our results indicated that the molecular weight of GPR17 was between 35 and 40 KDa, which was lower than the reported value. The molecular weight change of GPR17 protein in our study may be associated with GPR17 protein turnover[33]. However, to the best of our knowledge, there is no evidence to support the GPR17 protein turnover hypothesis.

New comment 2: It is unlikely that the change in molecular weight is related to protein turnover. It might be due to receptor post-translational modification (e.g., glycosylation). I recommend that the authors do not mention this sentence in the text. They should clearly label the molecular weight in the figures and describe the discrepancy as a different glycosylation state. This discrepancy does not necessarily relate to the reliability of their data since different cells may introduce different glycosylation.

Response: We thank the reviewer for the valuable advice. We labeled the molecular weight in the supplemental figures. We revised these sentence in the text as follows:

Changes in the text: The molecular weight change of GPR17 protein in our study might be due to receptor post-translational modification, such as glycosylation. The discrepancy between mRNA and protein results may be due to GPR17 protein turnover [33]. However, to the best of our knowledge, there is no literature evidence to support the GPR17 protein turnover. At present, this is stated only as a hypothesis. Further study is needed to verify whether the discrepancy between mRNA and protein results is caused by protein turnover.

Comment 3: The authors should also consider the protein turnover: is GPR17 turnover sufficiently fast to claim that the reduction of the transcript is relevant for cell functions? They should briefly comment this issue in the discussion.

Response: We thank the reviewer for pointing out it. We added a briefly comment about issue in the discussion as follows:

The molecular weight change of GPR17 protein in our study may be associated with GPR17 protein turnover[33]. However, to the best of our knowledge, there is no evidence to support the GPR17 protein turnover hypothesis.

New comment 3: See my previous comment. To me, this sentence makes no sense. If the authors do not trust their own data, because of the lack of reproducibility they should not comment on that. Otherwise, they should not exclude that discrepancy between mRNA and protein results are due to protein turnover. If there is no literature evidence about this hypothesis, it should be stated only as a hypothesis.

Response: We thank the reviewer for pointing out it. We revised it as follows:

Changes in the text: The discrepancy between mRNA and protein results may be due to GPR17 protein turnover [33]. However, to the best of our knowledge, there is no literature evidence to support the GPR17 protein turnover. At present, this is stated only as a hypothesis. Further study is needed to verify whether the discrepancy between mRNA and protein results is caused by protein turnover.

Comment 5: Figure 5e,f: The Y axis is not explained. Text and caption do not help. If the black bars in f are GPR17 with and without FOXO1, the results do not confirm FOXO1 binding. The authors should clarify. The binding between FOXO1 and GPR17 promoter has been demonstrated in very different experimental settings. Not necessarily this mechanism is present (or quantitatively relevant) in cell lines. Do the authors exclude that the effect on GPR17 is indirect?

Response: We thank the reviewer for pointing out it. We added the Y axis of Figure 5e and 5f in the manuscript.

New comment 5: The overall meaning of the “RATIO” should be explained. Why have the authors normalized vs. Normal Rabbit IgG? It may not be clear for those who are not familiar with the ChIP. Then, I suggest to simplify the label on the Y-axis, removing the product codes.

Response: We thank the reviewer for the valuable advice. We simplified the label and removed the product codes on the Y-axis of Figure 5e and 5f. We added an explanation of “ratio” in the legend of Figure 5 as follows:

Changes in the text: Data are presented as signal relative to FoxO1 (C29H4) Rabbit mAb #2880/Normal Rabbit IgG #2729 input ratio. Normal rabbit IgG #2729 is an internal reference. The ratio is used to determine the binding rate of FOXO1 to the promoter regions of GPR17.

Minor comments:

Comment 6: The first paragraph of the Introduction is misleading and should be removed since the manuscript is not about acute ischemic stroke.

Response: We thank the reviewer for pointing out it. [...] Therefore, we did not remove the first paragraph of the Introduction. We added an explanation to this in the Discussion section as follow:

To mimic the nerve injury under AIS, SK-N-SH cells were starved for 24 hours.

New comment 6: I see. In this case, I suggest that the authors add a brief mention also at the end of the introduction, otherwise, that paragraph seems off-topic.

Response: We thank the reviewer for the valuable advice. We added a brief mention about it in the end of the introduction.

Changes in the text: To mimic the nerve injury under AIS, SK-N-SH cells were starved for 24 hours. Here, we investigate the influence of IGF-1 on neuronal GPR17 expression as well as the underlying molecular mechanisms of this interaction.